# Letermovir Rescue Therapy in Kidney Transplant Recipients with Refractory/Resistant CMV Disease

**DOI:** 10.3390/jcm13010100

**Published:** 2023-12-23

**Authors:** Ellen von Hoerschelmann, Johannes Münch, Linde Gao, Christian Lücht, Marcel G. Naik, Danilo Schmidt, Paul Pitzinger, Detlef Michel, Parthenopi Avaniadi, Eva Schrezenmeier, Mira Choi, Fabian Halleck, Klemens Budde

**Affiliations:** 1Department of Nephrology and Medical Intensive Care, Charité Universitätsmedizin Berlin, 10117 Berlin, Germany; 2Institute of Virology, Charité Universitätsmedizin Berlin, Labor Berlin-Charité-Vivantes GmbH, 10117 Berlin, Germany; 3Institute of Virology, Universitätsklinikum Ulm, 89081 Ulm, Germany

**Keywords:** kidney transplantation, infection, cytomegalovirus, drug resistance

## Abstract

(1) Background: CMV infections remain a problem after kidney transplantation, particularly if patients are refractory or resistant (r/r) to treatment with valganciclovir (VGCV) or ganciclovir (GCV). (2) Methods: In a single-center retrospective study, kidney transplant recipients (KTR) receiving letermovir (LTV) as rescue therapy for VGCV-/GCV-r/r CMV disease were analyzed regarding CMV history, immunosuppression, and outcomes. (3) Results: Of 201 KTR treated for CMV between 2017 and 2022, 8 patients received LTV following treatment failure with VGCV/GCV. All patients received CMV prophylaxis with VGCV according to the center’s protocol, and 7/8 patients had a high-risk (D+/R−) CMV constellation. In seven of eight cases, rising CMV levels occurred during prophylaxis. In seven of eight patients, a mutation in UL97 associated with a decreased response to VGCV/GCV was detected. In four of eight patients, LTV resulted in CMV clearance after 24 ± 10 weeks (16–39 weeks), two of eight patients stabilized at viral loads <2000 cop/mL (6–20 weeks), and two of eight patients developed LTV resistance (range 8–10 weeks). (4) Conclusion: LTV, which is currently evaluated for CMV prophylaxis in kidney transplantation, also shows promising results for the treatment of patients with VGCV/GCV resistance despite the risk of developing LTV resistance. Additional studies are needed to further define its role in the treatment of patients with CMV resistance.

## 1. Introduction

CMV infection in kidney transplant recipients (KTR) continues to be a risk factor for graft loss and death and results in considerable morbidity [1]. Valganciclovir (VGCV) and ganciclovir (GCV) are the recommended agents of choice for CMV prophylaxis and treatment [2], and at-risk patients (D+/R−, D+/R+, D−/R+) generally receive universal prophylaxis with VGCV. However, approximately 25% of KTR develop CMV infection or disease [1]. Treatment is further complicated in cases of disease refractory or resistant (r/r) to VGCV/GCV, which occurs after at least 2 weeks of full-dose antiviral therapy without achieving a distinctive decrease in the CMV viral load. In these cases, resistance testing is advised [2]. Resistance-inducing mutations (M460V/I, H520Q, A594V, L595S, C603W, and C592G) appear most often in the UL97 gene, encoding viral kinases, followed by the UL54 gene, which encodes viral polymerase [2,3]. Risk factors for the development of resistance are a high-risk serostatus (D+/R−), an extended treatment period >5 months, intense immunosuppression, and insufficient bioavailability. Reduced VGCV dosage due to leukopenia is common in KTR, and the need for eGFR adjustment further complicates the correct dosing of VGCV/GCV [4]. Resistance itself is associated with a higher rate of rejection in KTR and higher morbidity and mortality [1,2,5].

Alternative therapy options in solid-organ transplantation include high-dose GCV, foscarnet, and the recently approved maribavir [2,6,7,8,9]. Invasive disease recommendations favor the use of foscarnet over maribavir because of its limited drug penetration [10]. Yet, foscarnet is known for its strong nephrotoxicity and therefore is seldom used in KTR [11].

In stem cell recipients, prophylaxis and treatment options include letermovir (LTV). LTV is a CMV-specific agent that targets the terminase enzyme complex and has shown a favorable safety profile [3,12,13]. It is currently being investigated for prophylaxis in KTR and is shown to be not inferior to VGCV [14]. In the need for alternative therapies in VGCV-/GCV-r/r disease, LTV has occasionally been used off-label, yet data to evaluate the role of LTV as an additional therapeutic option in KTR remain limited [15,16,17,18,19]. In this case study, we analyzed the use of LTV in 8 patients with VGCV-/CGV-r/r disease.

## 2. Materials and Methods

Patients/Cohort: We reviewed all patients who received a living or deceased donor kidney transplant at our center between 2017 and 2022 for CMV DNAemia or CMV disease. The identified patients were screened for VGCV/GCV resistance and for LTV treatment. Additional clinical and outcome data were collected from the electronic health record TBase [20]. The study was approved by the ethics committee of Charité Universitätsmedizin Berlin.

CMV Prophylaxis/Treatment Standard: CMV prophylaxis and treatment followed our center’s standard operation procedure. Patients and donors were screened for serological CMV status before transplant. In the case of a high-risk CMV constellation with the donor being CMV-IgG positive and the recipient CMV-IgG negative (D−/R+), prophylaxis with eGFR-adjusted VGCV lasting 6 months was initiated [4,21]. Patients with a moderate CMV risk (R+) received the prophylaxis for 3 months.

Virologic and Drug Testing: Patients were tested for systemic CMV replication regularly. Additional testing was performed when patients presented with possible clinical signs for CMV disease as fatigue, fever, infection of unknown origin, diarrhea, or leucopenia. The CMV DNA PCR as well as CMV resistance testing was provided by Labor Berlin (Berlin, Germany). The CMV DNA limit of detection was 300 cop/mL. Results between 300 and 2000 cop/mL are displayed as <2000 cop/mL. LTV resistance testing was performed by the CMV reference laboratory at Universitätsklinikum Ulm (Ulm, Germany).

Definition CMV DNAemia/CMV Disease: CMV DNAemia and CMV disease were defined, as described by Kotton et al. [2]. CMV DNAemia was described as systemic virus detection without specific CMV disease symptoms being present. CMV disease was defined as CMV DNA detection with attributable symptoms, such as fatigue, fever, diarrhea, or leukopenia.

## 3. Results

Between 2017 and 2022, 201 KTR were treated for CMV DNAemia or CMV disease at our center. A total of 8/201 patients received LTV as CMV treatment following treatment failure with VGCV/GCV. Figure 1 displays the patients’ viral loads and the treatment regimes.

All patients received standard immunosuppression at transplant, while none received intensified immunosuppression for AB0i transplant. One patient experienced steroid-refractory BANFF IIa/III rejection before the study incident, and another patient experienced BANFF IIa rejection during the incident, with both receiving intensified immunosuppression, including a methylprednisolone bolus and even anti-thymocyte globuline. Three patients received belatacept as primary immunosuppression before the studied event. Baseline demographics are provided in Table 1, additional laboratory data is provided in the Appendix A.

Seven of eight patients had a high-risk (D+/R−) CMV constellation. In seven of eight patients, a rising CMV viral load was detected under ongoing prophylaxis with VGCV (after 18 ± 7 weeks). Prophylaxis was temporarily interrupted due to leukopenia in one of these patients. One patient had CMV DNAemia, and seven of eight patients developed CMV disease. Two patients developed VGCV-/GCV-r/r CMV disease in the context of recurrent CMV disease.

Treatment with VGCV (*n* = 8) and intravenous GCV (*n* = 4) was intensified according to the manufacturer’s recommendations after detection of a rising CMV viral load. One patient received LTV as rescue therapy for VGCV-/GCV-refractory CMV disease, and no VGCV/GCV resistance was detected. In seven of eight patients, a mutation in the UL97 gene associated with a decreased response to VGCV/GCV was detected. Resistance testing was initiated during rising or refractory viral loads during VGCV/GCV therapy. The median VGCV/GCV treatment duration until the detection of resistance and switch to LTV was 11 weeks, with a wide range (3–35 weeks).

In four of eight patients, LTV resulted in CMV virus clearing after 24 ± 10 weeks (range 16–39 weeks) of treatment. In two additional patients, CMV stabilized at <2000 cop/mL (range 6–20 weeks). However, two of eight patients developed LTV resistance during a 2-month treatment period with LTV. Interestingly, both patients with LTV resistance showed restored sensitivity to VGCV/GCV.

In addition to the CMV treatment, immunosuppression was decreased in all patients to support CMV antiviral treatment. All three patients who received belatacept were switched back to a CNI-based regimen. MMF/MPA was decreased and eventually discontinued in seven of eight cases, while one patient did not receive antimetabolite treatment beforehand.

In two of eight patients, LTV was initially started in addition to VGCV treatment and VGCV was discontinued at a later time point. Furthermore, three of eight patients received CMV hyperimmunoglobulin, all in combination with VGCV/GCV. Only one patient received CMV hyperimmunoglobulin before detection of VGCV resistance. In two of three cases, CMV hyperimmunoglobulin was administered together with the reinstated VGCV/GCV treatment after detection of LTV resistance.

All six patients who were CMV-IgG negative before this studied incident completed seroconversion during the course of treatment.

### 3.1. Case 1

A 30-year-old male with end-stage renal disease (ESRD) caused by reflux nephropathy received a living donor kidney transplant in 2020. The induction immunosuppression consisted of basiliximab, as well as maintenance immunosuppression with tacrolimus, mycophenolate mofetil, and prednisolone. Immunosuppression was changed to belatacept 2 months after the transplant, due to biopsy-confirmed interstitial fibrosis and poor renal function. Due to a high-risk CMV constellation (D+/R−), he received eGFR-adjusted CMV prophylaxis with VGCV to last for 6 months. After 27 weeks of ongoing prophylaxis, the patient presented with leukopenia. CMV disease was detected, and VGCV at a therapeutic dosage was started. The MMF dosage was reduced and ultimately discontinued. VGCV treatment led to a decrease in CMV levels to <2000 cop/mL. However, CMV levels increased again after 32 weeks. Resistance testing was initiated at increasing CMV levels, and VGCV resistance with a C603W mutation in UL97 was detected. Consequently, treatment with LTV was initiated at 480 mg/d. VGCV treatment was continued for a total of 53 weeks, and treatment was switched to LTV monotherapy. CMV levels stabilized at <2000 cop/mL. After 11 weeks, LTV dosing was intensified to 960 mg/d to address a potentially low LTV resistance threshold, which was well tolerated. After a total of 30 weeks of LTV, CMV clearance was achieved and seroconversion was detected. LTV was discontinued after a total treatment duration of 34 weeks and consecutive negative CMV results. During treatment, immunosuppression with tacrolimus was reinstated and continued together with methylprednisolone. Following stable CMV clearing, MMF was reestablished at 500 mg/d. The eGFR before CMV disease was 39 mL/min, remained stable, and increased to 52 mL/min at the end of treatment.

### 3.2. Case 2

A 71-year-old female with ESRD caused by Goodpasture syndrome received a deceased donor kidney transplant in 2019. The induction immunosuppression consisted of basiliximab, as well as maintenance immunosuppression with tacrolimus, mycophenolate mofetil, and prednisolone. Due to a high-risk CMV constellation (D+/R−), she received eGFR-adjusted CMV prophylaxis with VGCV to last for 6 months. After 18 weeks of ongoing prophylaxis, the patient developed diarrhea and leukopenia. Additionally, the bloodwork showed grade 1 lymphopenia. CMV disease was detected. The VGCV dosage was increased to a therapeutic dose, but virus levels increased. A H520Q variant in the UL97 gene was detected, and VGCV was switched to LTV at a daily dose of 480 mg. After 6 weeks of LTV at a therapeutic dosage, CMV < 2000 cop/mL was achieved and the dosage was decreased to 480 mg every second day for another 11 weeks. During this time, CMV clearance was not achieved, yet CMV stabilized at <2000 cop/mL even without ongoing prophylaxis. During the course of CMV treatment, immunosuppression with MMF was decreased and discontinued until CMV < 2000 cop/mL was achieved. MMF was reinstated at a reduced daily dose of 1 g. Graft function was not altered during CMV disease or treatment. The eGFR before the CMV disease was 35 mL/min and at stop of prophylaxis was 36 mL/min.

### 3.3. Case 3

A 52-year-old male with ESRD caused by IgA nephropathy underwent a second kidney transplant in 2018. The induction immunosuppression consisted of basiliximab, as well as maintenance immunosuppression with tacrolimus, mycophenolate mofetil, and prednisolone. Due to a high-risk CMV constellation (D+/R−), he received eGFR-adjusted CMV prophylaxis with VGCV to last for 6 months. Due to leukopenia, VGCV had to be paused 6 weeks after the transplant for 3 weeks. Then, 23 weeks after transplantation, CMV disease was detected and initial GCV therapy was initiated. The viral load increased further after 4 weeks of treatment with GCV/VGCV. Resistance testing revealed an A594T mutation in the UL97 gene. Therapy was switched to LTV 240 mg/d, and CMV < 2000 cop/mL was achieved after a further 19 weeks. LTV therapy was stopped after 24 weeks, following two consecutive tests confirming CMV < 2000 cop/mL. At the initial CMV detection, bloodwork showed grade 1 lymphopenia. During the course of CMV treatment, immunosuppression with MMF was decreased and discontinued until CMV < 2000 cop/mL was achieved. MMF was then reinstated at 500 mg/d. Graft function remained stable under CMV disease and treatment at an eGFR of 22 mL/min.

### 3.4. Case 4

A 44-year-old male with ESRD caused by IgA nephropathy received a kidney transplant in 2021. The induction immunosuppression consisted of basiliximab, as well as maintenance immunosuppression with tacrolimus, mycophenolate mofetil, and prednisolone. Due to a high-risk CMV constellation (D+/R−), he received eGFR-adjusted CMV prophylaxis with VGCV to last for 6 months. Then, 9 weeks after transplantation, CMV disease was detected and VGCV was intensified to a therapeutic dosage. After an initial decrease in the viral load, increasing CMV levels were detected and resistance testing was initiated. A C603W variant in the UL97 gene was found. Additional therapy with 480 mg/d LTV was initiated after 14 weeks of VGCV. VGCV was continued for a total of 30 weeks. CMV cleared after 39 weeks of LTV, and LTV was terminated after 41 weeks after two consecutive negative CMV PCR tests. During the course of CMV treatment, immunosuppression with MMF was decreased and further on discontinued. After CMV clearing, MMF was reinstated at 1 g/d. Graft function improved during resolution of CMV (eGFR pre-CMV disease, 35 mL/min; eGFR post-CMV disease, 70 mL/min).

### 3.5. Case 5

A 74-year-old male with ESRD caused by an unknown kidney disease received a kidney transplant in 2016. The induction immunosuppression consisted of basiliximab, as well as maintenance immunosuppression with CSA, mycophenolate mofetil, and prednisolone. Due to a high-risk CMV constellation (D+/R−), he received eGFR-adjusted CMV prophylaxis with valganciclovir to last for 6 months. In the first 6 months after transplantation, the patient developed CMV disease, which was successfully treated with GCV and VGCV. Seroconversion was achieved. Immunosuppression was reduced to dual therapy with CSA and prednisolone. Then, 3 years after transplantation, a rising CMV viral load without any ongoing prophylaxis was detected and GCV therapy was started. As CMV levels did not decrease, additional therapy with CMV hyperimmunoglobulin was initiated. Resistance testing revealed mutations in L595F in the UL97 gene. Therapy was switched to 480 mg/d LTV after 3 weeks of GCV/VGCV therapy. After 16 weeks of LTV, CMV clearance was achieved and LTV was discontinued. Immunosuppression was already reduced to dual therapy with CSA and prednisolone before the studied incident and was not reduced further. Graft function stayed stable at an eGFR of 20 mL/min during CMV treatment.

### 3.6. Case 6

A 63-year-old male with ESRD caused by IgA nephropathy received a kidney transplant in 2022. The induction immunosuppression consisted of basiliximab, as well as maintenance immunosuppression with tacrolimus, mycophenolate mofetil, and prednisolone. Due to a high-risk CMV constellation (D+/R−), he received eGFR-adjusted CMV prophylaxis with VGCV to last for 6 months. Then, 8 weeks after the transplant, rising CMV DNAemia was detected and therapy with GCV for 4 days was initiated and continued with VGCV. CMV levels decreased to <2000 cop/mL, yet clearance could not be achieved. Resistance testing revealed an A594V mutation in the UL97 gene. LTV was initiated at 480 mg/d after 15 weeks of VGCV/GCV treatment. CMV levels remained at <2000 cop/mL, and LTV was discontinued after 8 weeks. During follow-up, CMV levels increased briefly but were cleared without further treatment. During the course of CMV treatment, immunosuppression with MMF was decreased and further on discontinued for 1 month. MMF was continued at 500 mg/d. Graft function stayed stable at an eGFR of 37 mL/min.

### 3.7. Case 7

A 53-year-old female with ESRD caused by reflux nephropathy received a kidney transplant in 2018. The induction immunosuppression consisted of basiliximab, as well as maintenance immunosuppression with tacrolimus, mycophenolate mofetil, and prednisolone. Tacrolimus was switched to belatacept due to poor kidney function. The biopsy showed severe donor-dependent arteriosclerosis. Due to a high-risk CMV constellation (D+/R−), she received eGFR-adjusted CMV prophylaxis with VGCV to last for 6 months. Then, 23 weeks after the transplant, CMV disease was detected, and VGCV was intensified to a therapeutic dosage and further on switched to intravenous GCV for 13 days. Due to persistently rising CMV levels, therapy was switched to 480 mg/d LTV as rescue therapy. After 7 weeks of LTV and rising CMV levels, resistance testing was initiated, which showed mutations in C325Y and C325F in the UL56 gene, leading to LTV resistance. No mutations in resistance genes regarding alternative therapies, such as GCV, were found. Therapy with VGCV was re-started after 10 weeks of LTV, along with CMV hyperimmunoglobulin administration.

Parallel to CMV disease, the patient suffered from transplant pyelonephritis and showed acute BANFF IIa cellular rejection and partial necrosis in the kidney cortex on the kidney transplant biopsy, which led to acute kidney transplant injury requiring hemodialysis. At the detection of CMV disease, grade 2 lymphopenia was present. During the course of CMV treatment, immunosuppression was decreased with MMF dosage reduction and finally discontinued. Tacrolimus was reinstated instead of belatacept because of rejection. Unfortunately, the patient developed an abscess on the graft, and transplantectomy was performed without recovery of the graft function.

### 3.8. Case 8

A 52-year-old female with ESRD of unknown origin received a kidney transplant in 2018. The induction immunosuppression consisted of basiliximab, as well as maintenance immunosuppression with tacrolimus, mycophenolate mofetil, and prednisolone. Tacrolimus was changed to belatacept 3 months post-transplant, due to tacrolimus intoxication associated with thrombotic microangiopathy. Shortly afterward, the patient showed an acute kidney injury and the kidney transplant biopsy findings showed BANFF IIa/III rejection, which was treated with a 5-day methylprednisolone bolus and additional ATG. She received 3-month prophylaxis with VGCV (D−/R+). After completing prophylaxis, she developed CMV disease 10 months after the transplant, which was treated with 10 days of GCV and 6 weeks of VGCV, leading to CMV clearing. However, CMV reoccurred under ongoing prophylaxis 6 months after being cleared at first. Therapy with VGCV was reinitiated, but CMV levels continued to rise, and resistance testing showed an A594V mutation in the UL97 gene. LTV at a dose of 480 mg/d was started after 15 weeks of VGCV treatment, and the viral load persisted under LTV treatment for 10 weeks. Further resistance analysis showed a C325Y mutation in the UL56 gene, leading to LTV resistance. Yet, analysis showed restored GCV sensitivity. LTV was discontinued. CMV clearance was achieved with resumption of VGCV treatment and CMV hyperimmunoglobulin. At CMV detection, grade 3 lymphopenia was present. During CMV treatment, immunosuppression with CNI was reinstated and MMF was discontinued. Additionally, the patient developed absolute immunoglobulin deficiency after rejection therapy and received immunoglobulin twice during CMV treatment. Graft function remained stable at an eGFR of 50 mL/min.

## 4. Discussion

Discussions regarding alternatives to CMV therapy for KTR have become even more apparent with recent studies of LTV as a potential candidate for CMV prophylaxis [13] and the approval of maribavir for CMV-resistant refractory patients in 2022 [22]. Stoelben et al. already showed a proof of principle also for LTV as a CMV treatment option [12]. Yet, data regarding the use of LTV beyond prophylaxis in KTR remain limited, even if its safety profile and completely different mode of action make it an attractive alternative also in CMV treatment [19,23,24]. Here, we presented the largest case series of KTR treated with LTV in r/r CMV infection until now.

Our case series demonstrates that LTV is an effective treatment option in cases of VGCV/GCV resistance, and it led to sustained reduced viral loads <2000 cop/mL or clearing in six of eight cases, supporting further studies beyond prophylaxis. In seven of eight cases, graft function remained stable or even increased after successful CMV treatment. Unfortunately, one patient suffered graft loss unrelated to LTV treatment in a complex situation with complicated transplant pyelonephritis, leading to transplant abscess, rejection, and CMV disease.

In our case series, LTV was mostly used as monotherapy, and two patients who were started on a combination of VGCV and LTV were switched to monotherapy later. CMV hyperimmunoglobulin was only used prior to the switch to LTV. At the time of this case series, maribavir was not available, and alternatives, such as foscarnet or cidofovir, clearly have an inferior safety profile, especially for patients with already poor kidney function. In our series, seven of eight patients had an eGFR lower than 40 mL/min. LTV was used as a secondary prophylaxis in only one case at our center, and the overall goal was to end LTV treatment as soon as possible to limit the development of resistance to this expensive therapy. It is important to note that two of eight patients developed LTV resistance but simultaneously lost GCV resistance, allowing reintroduction of VGCV and clearance of CMV. In summary, our series demonstrates the feasibility, efficacy, and safety of LTV rescue treatment in patients r/r to VGCV/GCV.

Comparable data regarding the use of LTV in combination with other agents, such as VGCV/GCV or even CMV hyperimmunoglobulin, are scarce and show mixed results. In Jorgenson et al.’s study, no patient cleared to negativity with the addition of letermovir to VGCV, even though the combination should logically address the different CMV subtypes in r/r CMV disease [18]. In the population studied by Rho et al., all patients receiving LTV in combination with VGCV as a step-down therapy experienced a viral breakthrough [25].

Risk factors for VGCV/GCV resistance development in our cases included intensified immunosuppression, eGFR-adjusted decreased dosage of VGCV, VGCV prophylaxis, and a treatment duration of >5 months [2]. Interestingly, seven of eight patients developed CMV viremia during prophylaxis with VGCV in our series. A similar observation was made in a large prospective randomized trial, in which protocol-driven close surveillance detected a few GCV-resistant cases under prophylaxis with VGCV. More data are needed to better understand the rare development of resistance during VGCV prophylaxis.

Three of eight patients received belatacept prior to CMV detection. Belatacept has been shown to increase the risk of CMV infections and an atypical course of disease [26]. Underlying immunological mechanisms are not fully understood and may be related to the severely impaired humoral response with belatacept [27,28]. Furthermore, the inhibition of the naive T-cell response through belatacept may impair the immune response to primary CMV infection [26,29]. Interestingly, two of three of the belatacept-treated patients developed LTV resistance later. These two patients clearly had an impaired immune response, with one having received ATG and developing an absolute immunoglobulin deficiency after prednisolone-refractory rejection and the other receiving rejection treatment with a methylprednisolone bolus and experiencing complex urosepsis in parallel.

It is often emphasized that LTV has a low resistance threshold and the underlying mutations in UL56 often lead to absolute resistance [30,31]. LTV resistance is especially associated with its use as secondary prophylaxis or treatment [13,23,32]. Here, we reported LTV resistance during treatment in two of eight cases under massive immunosuppression, as discussed earlier. It is important to note that both patients had restored VGCV/GCV sensitivity, which is consistent with the finding that cross-resistance against LTV and VGCV/GCV does not occur [33]. The patient receiving longer-lasting secondary prophylaxis with LTV showed no signs of resistance, as well as the patients who received LTV for a longer period (24–41 weeks).

A limitation of our case series is the small number of patients, each experiencing a complex clinical course after transplantation. Despite this limitation, our case series demonstrates the potential of LTV as an alternative effective and well-tolerated CMV treatment in the case of the rare VGCV-/GCV-r/r cases. With the approval of maribavir, another effective and safe alternative treatment is available for such cases. More data are needed to determine the best treatment regime under such circumstances and whether LTV should be used as monotherapy or in combination with other drugs. Most importantly, our case series may help other VGCV-/GCV-r/r patients to find an effective and non-nephrotoxic treatment and provide a rationale for further investigations.

## Figures and Tables

**Figure 1 jcm-13-00100-f001:**
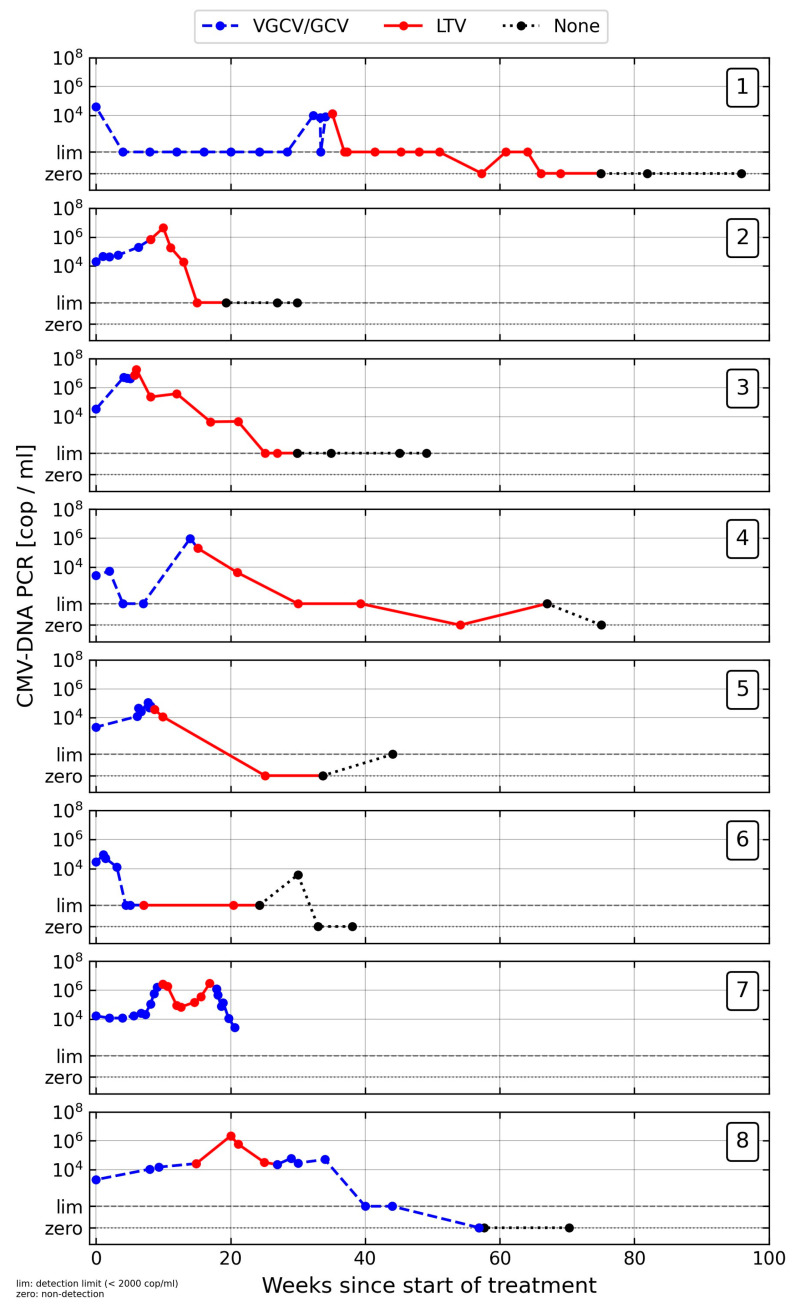
Timeline of CMV viral load and treatment. CMV viral loads at weeks since the start of CMV treatment are displayed for cases 1–8 individually. The colors indicate the type of treatment received by the patient at the time (blue = VGCV/GCV, red = LTV, black = none).

**Table 1 jcm-13-00100-t001:** Patient characteristics.

GFR after CMV	Best GFR before CMV	LTV Resistance	Additional/Parallel Therapy		LTV to Clearing (w)	LTV to <2000 cop/mL (w)	LTV (w)	VGCV/GCV Resistance	VGCV/GCV Treatment to LTV (w)	VGCV Prophylaxis Discontinued	CMV under Prophylaxis	CMV > 2000 cop/mL after Tx (w)	CMV Infection	Serostatus Prior to CMV	Rejection	IS Prior to r/r CMV	Intensified IS	DD/LD	Initial IS	Age at Tx (y)	Sex	Case
50 mL/min	38 mL/min	No	Yes, LTV + VGCV		30.7	1.4	34.4	Yes, UL97:C603W	35.1	No	Yes	27.0	1°	D+/R−	No	Belatacept, MMF 2 g/d, Urbason	No	LD	Basiliximab, CNI, MMF, Urbason	30	M	1
36 mL/min	30 mL/min	No	No		No clearing	6.0	6.0	Yes, UL97:H520Q	9.1	No	Yes	18.0	1°	D+/R−	No	CNI, MMF 2 g/d, Urbason	No	DD	Basiliximab, CNI, MMF, Urbason	71	W	2
20 mL/min	16 mL/min	No	No		No clearing	19.7	24.4	Yes, UL97:A594T	4.7	No	Yes	18.4	1°	D+/R−	No	CNI, MMF 1 g/d, Urbason	No	DD	Basiliximab, CNI MMF, Urbason	52	M	3
70 mL/min	25 mL/min	No	Yes, LTV + VGCV		38.9	14.7	41.1	Yes, UL97:C603W	13.1	No	Yes	9.4	1°	D+/R−	No	CNI, MMF 2 g/d, Urbason	No	DD	Basiliximab, CNI, MMF, Urbason	44	M	4
18 mL/min	20 mL/min	No	Yes, VGCV +	Cytotect	16.3	-	16.4	Yes, UL97:H469Y, L595F	2.7	Yes, 3 w	No	274.6	2°	D+/R−	No	CSA, Urbason	No	DD	Basiliximab, CSA, MMF, Urbason	74	M	5
36 mL/min	40 mL/min	No	No		16.7	-	8.1	Yes, UL97:A594V	14.9	No	Yes	8.3	1°	D+/R−	No	CNI, MMF 1 g/d, Urbason	No	DD	Basiliximab, CNI, MMF, Urbason	63	M	6
<15 mL/min	<15 mL/min	Yes, UL56:C325Y, C325F	Yes, VGCV + Cytotect		-	-	7.6	No	9.3	No	Yes	22.6	1°	D+/R−	Yes	Belatacept, MMF 2 g/d, Urbason	Yes, Methylprednisolone bolus	DD	Basiliximab, CNI, MMF, Urbason	53	W	7
60 mL/min	51 mL/min	Yes	Yes, VGCV + Cytotect		-	-	9.7	Yes, UL97:A594V	15.1	No	Yes	79.7	Recurrent infection	D−R+	Yes	Belatacept, MMF 1 g/d, Urbason	Yes, Methylprednisolone bolus, ATG	DD	Basiliximab, CNI, MMF, Urbason	52	W	8

## Data Availability

The data presented in this study are available on request from the corresponding author. The data are not publicly available due to privacy reasons.

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
