# Peer review of "Letermovir Rescue Therapy in Kidney Transplant Recipients with Refractory/Resistant CMV Disease"

_jcm, 2023, doi:10.3390/jcm13010100_

Round 1
Reviewer 1 Report
Comments and Suggestions for Authors
Dear Authors,
Thank you for your manuscript entitled "Letermovir Rescue Therapy in Kidney Transplant Recipients with Refractory/Resistant CMV Disease"
I have few comments that I wish that may help in supporting your manuscript:
1- Since the main objective of this work is estimate the efficacy of Letermovir as "CMV terminase inhibitor",
I wish that the authors would support with more clinical data like total leukocytic count with lymphocyte differential or kidney/liver function tests and HLA matching tests.
These data will help to make the study more solid.
2- it is not clear how you do measure the graft rejection in the article.
3- CMV viral titration is measured by VIral DNA copies, is there any measurement of other parameters to confirm? like Donor specific antibodies, or protein electrophoresis to show antibodies, et..
Author Response
Thank you for reviewing our manuscript. Please find our response to your comments below:
1: If leucopenia occurred in the beginning or during the course of the CMV disease and its treatment it is mentioned in the case description. Data on lymphocyte count, if available, has been supplemented to the manuscript.
2: Both patients who were diagnosed with a rejection received a kidney transplant biopsy to determine the cause of the acute kidney injury. Graft rejection was diagnosed and classified according to BANFF via nephropathological assessment of the taken biopsy.
3: Viral load was assessed via the measurement of CMV DNA copies. Additional to the viral load the CMV IgG negative patients were tested for seroconversion regularly.
Reviewer 2 Report
Comments and Suggestions for Authors
Dear Authors,
Besides limitations regarding the small number of patients, you have described a series of cases well, and in my opinion, your results can arouse your interest and be a guideline for further determining the importance and potential of LTV as an alternative effective and well- tolerated CMV 342 treatment in case of the rare VGCV/GCV r/r cases. For clinicians it is important to reveal more details as interaction potential and dosage optimization methods for transplant patients.
Thank you for this article,
Sincerely,
Author Response
Thank you for your feedback on our manuscript. We agree that more data and studies including more patients regarding the treatment regime with letermovir, especially regarding dosage and possible combination therapy, is needed.
Round 2
Reviewer 1 Report
Comments and Suggestions for Authors
Dear Authors,
Thank you for adding some important edits to the manuscript. The paper looks better.
I do not see the supplementary data of the clinical data of the patients. Please be sure to be added.
Good luck.
Author Response
Thank you for your feedback. We have included the patients laboratory data to the supplementary files.
Kind regards,
Ellen von Hoerschelmann